# Investigation of α-Glucosidase Inhibitory Metabolites from *Tetracera scandens* Leaves by GC–MS Metabolite Profiling and Docking Studies

**DOI:** 10.3390/biom10020287

**Published:** 2020-02-12

**Authors:** Ahmed Nokhala, Mohammad Jamshed Siddiqui, Qamar Uddin Ahmed, Muhammad Safwan Ahamad Bustamam, Zainul Amiruddin Zakaria

**Affiliations:** 1Department of Pharmaceutical Chemistry, Kulliyyah of Pharmacy, International Islamic University Malaysia, Indera Mahkota, Kuantan 25200, Pahang, Malaysia; ph.ahmednokhala@gmail.com (A.N.); quahmed@iium.edu.my (Q.U.A.); 2Pharmacognosy Research Group, Kulliyyah of Pharmacy, International Islamic University Malaysia, Indera Mahkota, Kuantan 25200, Pahang, Malaysia; 3Laboratory of Natural Products, Institute of Bioscience, Universiti Putra Malaysia, Serdang 43400, Selangor, Malaysia; safwan.upm@gmail.com; 4Department of Biomedical Sciences, Faculty of Medicine and Health Sciences, Universiti Putra Malaysia, Serdang 43400, Selangor, Malaysia; 5Halal Research Institute, Universiti Putra Malaysia, Serdang 43400, Selangor, Malaysia

**Keywords:** *Tetracera scandens*, metabolite profiling, α-glucosidase inhibition, orthogonal partial least squares, GC–MS metabolomics, molecular docking

## Abstract

Stone leaf (*Tetracera scandens*) is a Southeast Asian medicinal plant that has been traditionally used for the management of diabetes mellitus. The underlying mechanisms of the antidiabetic activity have not been fully explored yet. Hence, this study aimed to evaluate the α-glucosidase inhibitory potential of the hydromethanolic extracts of *T. scandens* leaves and to characterize the metabolites responsible for such activity through gas chromatography–mass spectrometry (GC–MS) metabolomics. Crude hydromethanolic extracts of different strengths were prepared and in vitro assayed for α-glucosidase inhibition. GC–MS analysis was further carried out and the mass spectral data were correlated to the corresponding α-glucosidase inhibitory IC_50_ values via an orthogonal partial least squares (OPLS) model. The 100%, 80%, 60% and 40% methanol extracts displayed potent α-glucosidase inhibitory potentials. Moreover, the established model identified 16 metabolites to be responsible for the α-glucosidase inhibitory activity of *T. scandens*. The putative α-glucosidase inhibitory metabolites showed moderate to high affinities (binding energies of −5.9 to −9.8 kcal/mol) upon docking into the active site of *Saccharomyces cerevisiae* isomaltase. To sum up, an OPLS model was developed as a rapid method to characterize the α-glucosidase inhibitory metabolites existing in the hydromethanolic extracts of *T. scandens* leaves based on GC–MS metabolite profiling.

## 1. Introduction

Omics technologies, including genomics, transcriptomics and proteomics, aim to give a holistic view about a biological system by studying different subcellular components (i.e., DNA, RNA and protein respectively) [1]. Metabolomics is a relatively new member of the “omics” group that studies the whole set of low molecular weight compounds (the metabolome) existing in a biological sample [2]. The great advancements of sequencing techniques, high-throughput analytical platforms and chemometric tools that have taken place over the past few decades stood behind the emergence and prosperity of the omics technologies [3]. Recently, metabolomics has become a routine approach in studies aiming to evaluate the activities of medicinal plants or to characterize the metabolites responsible for such activities [4,5,6,7]. Classical methods for determination of the active plant metabolites rely on the activity-guided fractionation and purification techniques. However, this process usually takes a long time and may end up with the isolation of few known metabolites with already known activities [8]. By contrast, metabolomics studies consider almost all the existing metabolites in a rapid analysis, then the meaningful correlations with the activity are mined through multivariate statistical analysis. The high-throughput nature makes the metabolomics approach more suitable to fulfill the high demands of the modern drug discovery policies, hoping to bring the natural products into the spotlight again as a potential source of new drug leads [2]. 

Metabolic fingerprinting and metabolite profiling are the main strategies followed in plant metabolomics studies. The former uses non-specific analytical platforms to obtain a spectral fingerprint of all metabolites present in the sample (no information is gained about the identity of those metabolites), while the latter identifies (and sometimes quantifies) a pre-defined set of related metabolites, such as those involved in a specific metabolic pathway or those responsible for certain bioactivity [9]. Mass spectrometry has been used successfully for both strategies. An unhyphenated mass spectrometer is a universal detection tool that has been used for data acquisition in metabolic fingerprinting studies. Moreover, when attached to a chromatographic unit (e.g., gas chromatograph, high performance liquid chromatograph or ultra performance liquid chromatograph), mass spectrometry can be used for metabolite profiling purposes. In such a case, the metabolites are introduced into the ionization chamber sequentially instead of concomitantly, enhancing the quality of the acquired mass spectral data (i.e., better sensitivity and reduced background noise) leading to better qualitative and quantitative results [10]. GC–MS-based metabolomics has been used for profiling of the active metabolites from *Clinacanthus nutans* [5], *Cosmos caudatus* [11], *Camellia sinensis* [12], *Achras sapota* [13], and from selected commercial essential oils [14]. Figure 1 summarizes the procedure followed in the aforementioned studies. Generally, it involves the GC–MS analysis as well as the assessment of the activity of the different samples. The datasets acquired during metabolomics studies are usually complex (i.e., have multiple variables and observations), hence they require sophisticated statistical tools for their analysis; namely multivariate data analysis (MVDA). MVDA tools rely on the concept of dimensionality reduction to mine the meaningful correlations between the X-variable (e.g., mass spectral data) and the Y-variable (e.g., bioactivity). Partial least squares (PLS) is an MVDA tool that forms new X-variables (referred to as latent variables) that are linear combinations of the old ones and give a good correlation with the Y-variable at the same time. The first PLS component describes more variation of the X-variables and predicts more variation of the Y-variable than the second PLS component, and so on. An orthogonal partial least squares (OPLS) algorithm is an extension of the PLS algorithm that separates the variation in X into two parts, one that is related to Y (called predictive component) and one that is unrelated to Y (called orthogonal components) [15]. The active metabolites are further identified from the loading column plot of the developed multivariate model, where their representative columns are shown to be positively correlated with the activity. Murugesu et al. reported the metabolites namely; palmitic acid, 1-monopalmitin, pentadecanoic acid, hexadecanoic acid, heptadecanoic acid, phytol, stigmasterol, stigmast-5-ene, glycerol monostearate, 1-linolenoyl glycerol and alpha-tocospiro B to be responsible for the α-glucosidase inhibitory activity of *C. nutans* leaves extracts. On molecular docking, these putative active metabolites displayed predicted binding energies of −3.75 to −9.09 Kcal/mol, with stigmast-5-ene, stigmasterol and alpha-tocospiro B showing better affinities compared to the control ligand and the positive control, quercetin [5]. Similarly, Javadi et al. reported the metabolites namely; catechin, α-d-glucopyranoside, α-tocopherol and α-linolenic acid to elicit the α-glucosidase inhibitory activity of *C. caudatus* leaves extracts [11]. Furthermore, Das et al. reported that chlorogenic acid and gallic acid were responsible for the α-glucosidase inhibitory activity of *A. sapota* fruit extracts [13]. 

*Tetracera scandens* is a Southeast Asian medicinal plant with several indications, including diabetes mellitus, hepatitis, rheumatism, urinary illnesses, hypertension and diarrhea [16]. Muliyah et al. have demonstrated the antibacterial potential of the methanol extract of *T. scandens* stem as a proposed mechanism of the antidiarrheal activity of the plant [17]. Moreover, an in vivo study has been conducted to investigate the hepatoprotective and the antioxidant potential of the ethanol extract of *T. scandens* leaves. The results of this study showed that *T. scandens* has a considerable protective effect against carbon tetrachloride- induced hepatotoxicity in a rat model, which may support the traditional use of the plant for the management of hepatitis [18]. In a large screening study of nearly 100 plant species, the xanthine oxidase inhibitory activity has been reported for the hydromethanolic extract of *T. scandens* for the first time [19]. Later on, the activity-guided fractionations and chromatographic purifications have led to the isolation of six compounds with significant xanthine oxidase inhibitory activity, namely platanic acid, 28-*O*-*β*-d-glucopyranosyl ester of platanic acid, betulinic acid, tiliroside, kaempferol and quercetin [20]. Furthermore, the antidiabetic activity of the water and methanol extracts of *T. scandens* leaves has been verified in an in vivo study using an alloxan-induced diabetic rat model [16]. Moreover, Lee et al. have investigated the glucose-uptake enhancing potential of different partitions of the methanol extract of a *T. scandens* branch. The flavonoids 3’,5’-diprenylgenistein, 6,8-diprenylgenistein, alpinumisoflavone and derrone were isolated from the active ethyl acetate partition, and were reported to be responsible for such activity [21]. This study aimed to evaluate the α-glucosidase inhibitory potential of the hydromethanolic extracts of *T. scandens* leaves and to characterize the metabolites responsible for such activity through GC–MS metabolomics.

## 2. Materials and Methods 

### 2.1. Materials 

Methanol and pyridine were supplied from Merck (Darmstadt, Germany). α-Glucosidase (from *Saccharomyces cerevisiae*) was purchased from Megazyme (Wicklow, Ireland). Potassium dihydrogen phosphate, *p*-nitrophenyl-α-d-glucopyranoside (PNPG), methoxyamine hydrochloride, quercetin standard and *N*-methyl-*N*-(trimethylsilyl) trifluoroacetamide (MSTFA) were purchased from Sigma-Aldrich (St. Louis, MO, USA). Glycine was procured from Nacalai Tesque Inc. (Kyoto, Japan). Dimethyl sulfoxide (DMSO) was obtained from Fisher scientific UK (Loughborough, Leics, UK).

### 2.2. Plant Material

The leaves of *T. scandens* were freshly collected from the forest at Tasik Chini, Pahang, Malaysia. The leaves were washed with water to get rid of any debris or microbial growth and initially dried with tissue paper. A sample was deposited at the herbarium of Kulliyyah of Pharmacy, International Islamic University Malaysia (voucher specimen no. PIIUM 0305).

### 2.3. Sample Preparation

The leaves of *T. scandens* were dried at room temperature for 10 days, ground into coarse powder using a laboratory blender and stored at −20 °C till extraction. A total of 36 extracts were prepared using methanol of different strengths (100%, 80%, 60%, 40%, 20%, and 0%). Approximately 150 mL of the solvent was added to about 10 g of the powdered leaves with sonication for 30 min. After filtration through a Büchner funnel, the liquid phase was introduced into a rotary vacuum evaporator (Büchi^®^, Flawil, Switzerland) to evaporate the solvent at 40 °C. Furthermore, the remaining moisture was eliminated via freeze drying (Alpha 1–2 LD plus, Martin Christ, Osterode, Germany). The crude extracts were maintained at −80 °C until analyzed. The extraction yield was calculated using the following equation [11,22]: Extraction yield (%, *w*/*w*) = Weight of the crude extract/Weight of the plant raw material × 100(1)

### 2.4. Assay of α-Glucosidase Inhibitory Activity

The method reported by Saleh et al. [23] was followed for assessment of the α-glucosidase inhibitory potential of *T. scandens* extracts, with slight modifications. Quercetin (positive control) and the lyophilized plant extracts were dissolved in DMSO to obtain stock solutions of concentrations 2 and 4 mg/mL, respectively. One hundred µL of the phosphate buffer (30 mM, pH 6.5) was added to each well. Afterward, 10 µL of the serially diluted extract solutions was added to the sample wells (to get final concentrations of 160, 80, 40, 20, 10 and 5 µg/mL). Similarly, 10 µL of DMSO and quercetin solution were added to the control and positive control wells, respectively. Furthermore, 15 µL of the freshly prepared enzyme solution in phosphate buffer (50 mM, pH 6.5) was added to the sample, control and positive control test wells (0.03 U/well). An equal amount of the phosphate buffer (50 mM, pH 6.5) was added to the sample, control and positive control blank wells. The mixture was allowed to incubate for 5 min at room temperature, then 75 µL of the substrate solution (0.3 mg/mL PNPG in 50 mM phosphate buffer pH 6.5) was added to all wells and incubated for a further 15 min. Thereafter, 50 µL of glycine solution (2M, pH 10) was added to stop the reaction and the absorbance was measured using a microplate reader (NanoQuant Infinite M 200, Tecan, Grodig, Austria) at 405 nm. The absorbances of blank wells were subtracted from their corresponding test well absorbances, then the percent of α-glucosidase inhibition was calculated according to the equation given below and the α-glucosidase inhibitory IC_50_ value for each sample was calculated from the regression line between sample concentration (µg/mL) on the X axis and % inhibition on the Y axis.
Inhibition (%) = [(A_control_ − A_sample_)/A_control_] × 100(2)

### 2.5. Gas Chromatography–Mass Spectrometry (GC–MS) Analysis

#### 2.5.1. Derivatization Procedure

A two-step procedure was followed for derivatization of the plant extracts before carrying out the GC–MS analysis as reported by Robinson et al. [24], with some modifications. Initially, 50 µL pyridine was mixed with approximately 10 mg of the freeze-dried *T. scandens* extract and sonicated for 5 min. Thereafter, 100 µL of 20 mg/mL methoxyamine hydrochloride in pyridine was added to the mixture and incubated in an incubator shaker (Innova 4000-M 1192, Weender Landstr, Goettingen, Germany) at 60 °C and 100 rpm for 2 h. Afterward, 300 µL of MSTFA was added to the mixture and further incubated for 30 min in the same conditions. Finally, the derivatized sample was filtered through a 0.45 µm syringe filter, covered with aluminum foil and allowed to stand overnight at room temperature to ensure completion of the reaction.

#### 2.5.2. GC–MS Analysis Conditions

The GC–MS analysis was performed following the procedure reported by Murugesu et al. [5], with some modifications. One microliter of the derivatized samples was injected in the splitless mode into a GC–MS system, consisting of an Agilent 6890 gas chromatograph and an HP 5973 mass selective detector (Agilent Technologies, Santa Clara, CA, USA). The extracts were separated on a DB-5MS 5% phenyl methyl siloxane column with an inner diameter (ID) of 250 µm and a film thickness of 0.25 µm (Agilent Technologies, Santa Clara, CA, USA) using helium as the carrier gas at a flow rate of 1 mL/min. The initial oven temperature was set to 100 °C for 5 min, and then increased sequentially to a target temperature of 190 °C at a rate of 10 °C/min, then to 204 °C at a rate of 1 °C/min and eventually to 325 °C at a rate of 2 °C/min with a total run time of 88.5 min. The injector and ion source temperatures were set to 250 °C and 230 °C, respectively. Mass spectra were acquired using a full scan mode with a mass range of 50 to 550 amu. The detector was set to a solvent delay of 6 min.

### 2.6. Data Preprocessing and Statistical Analysis

The raw mass spectral data (D files) obtained from Agilent Chemstation was first converted into the universal cdf format using ACD/Spectrus processor v 14.00 (Advanced Chemistry Development, Inc., ACD/Labs Ontario, Toronto, ON, Canada). The cdf files of the 6 extract groups were further imported into XCMS software (R version, The Scripps Research Institute, San Diego, CA, USA) for comprehensive preprocessing, involving feature detection, retention time correction, peak grouping, and alignment. Moreover, the measured IC_50_ values of the 36 extracts were analyzed using Minitab 18 (Minitab Inc., State College, PA, USA) by one-way analysis of variance (ANOVA) with Tukey’s comparison test at 95% confidence interval. Afterward, the extracted ions data was pooled together with the α-glucosidase inhibitory IC_50_ values into a Microsoft Excel sheet and were imported into SIMCA 14.1 (Umetrics, Umeå, Sweden) for multivariate analysis, as X and Y variables respectively. After being autoscaled (unit variance scaling), an OPLS model was developed to correlate the mass spectral data of the plant extracts and their corresponding α-glucosidase inhibitory activities. The identity of the metabolites of interest was determined through comparing their mass spectra with those stacked in the National Institute of Standards and Technology library (NIST14) with a matching threshold of 70%, using the Automated Mass Spectral Deconvolution and Identification System (AMDIS) (Version 2.70, National Institute of Standards and Technology, Gaithersburg, MD, USA) and Agilent’s Deconvoluted Reporting Software (DRS) (Agilent Technologies, Santa Clara, CA, USA) [5,11].

### 2.7. Molecular Docking

The 3D structures (sdf files) of the putative active metabolites, as well as quercetin (positive control), were obtained from PubChem (https://pubchem.ncbi.nlm.nih.gov/). The crystal structure of maltose–*Saccharomyces cerevisiae* isomaltase complex (pdb file) was downloaded from Protein Data Bank (http://www.rcsb.org/) (PDB ID: 3A4A) [5,25,26]. The protein (pdb) file was first processed by AutoDock Tools 1.5.6 (The Scripps Research Institute, La Jolla, CA, USA) to remove water molecules and to add hydrogen atoms [27]. The processed pdb file was separated into two pdb files, one for the enzyme (isomaltase) and one for the ligand (maltose). The enzyme’s file was further readjusted (in terms of the added hydrogens and the assigned atomic charges) according to the pH value used during the in vitro α-glucosidase inhibition assay (i.e., pH 6.5) using the PDB2PQR server (http://nbcr-222.ucsd.edu/pdb2pqr_2.0.0/) (National Biomedical Computation Resource, San Diego, CA, USA). The resulting pqr file was converted again to the pdb format using PyMol software (V 1.7.4, Delano Scientific, San Carlos, CA, USA), then the metal (Ca) line was added. Thereafter, the pdb files of both the ligand and the enzyme were converted into pdbqt format, the format needed for running the molecular docking job on AutoDock Vina. The docking method was validated via control docking of the experimentally bound ligand (i.e., maltose) before proceeding to dock the putative active metabolites [5,26]. The residues Asp215, Glu277 and Asp352 were reported to form the active site of the enzyme [25]. The control docking was repeated until the dimensions of the grid box were optimized. Eventually, the docking grid box was centered on the macromolecule (X, Y and Z coordinates of 21.284, −0.761 and 18.638 respectively) and its dimensions were set at 28 Å X 28 Å X 28 Å with a spacing of 1 angstrom. The docking job was performed with AutoDock Vina and the interactions in the ligand–enzyme complexes were visualized using LigPlot+ v.1.4.5 (European Bioinformatics Institute, Hinxton, Cambridge, UK). (Please see the Appendix A for detailed information.)

## 3. Results and Discussion

### 3.1. Extraction Yield and α-Glucosidase Inhibitory Activity

The extraction yield (%) and the α-glucosidase inhibitory IC_50_ values (µg/mL) of the six extract groups of *T. scandens* leaves are displayed in Figure 2A,B, respectively. The maximum yield was attained from the 20% methanol (14.60 ± 0.07%), while the minimum yield was obtained from the 40% methanol (11.34 ± 0.22%). Overall, the extraction yields from *T. scandens* leaves followed the trend 20% > 0% > 80% > 100% > 60% > 40% methanol. On the other hand, the α-glucosidase inhibitory activity followed a different trend, with the lowest potential attributed to the water extract (IC_50_ = 95.32 ± 2.32 µg/mL) followed by the 20% methanol extract (IC_50_ = 56.56 ± 1.68 µg/mL), while the highest activity (IC_50_ = 19.54 ± 2.5 µg/mL to 23.46 ± 3.91 µg/mL) was exhibited by the 40%, 60%, 80% and 100% methanol extracts, which were not significantly different from each other at *p* < 0.05. These results are consistent with [11,23,28] who have reported that the higher α-glucosidase inhibitory activity was associated with the higher alcohol to water ratios of the extractant mixtures, and vice versa.

### 3.2. Multivariate Data Analysis

Representative GC–MS chromatograms of the methanol and water extracts of *T. scandens* leaves are displayed in Figure 3. Correlation of the mass spectral data (X variable = 2689 features) and the α-glucosidase inhibitory IC_50_ values (Y variable = 1) collected from 36 extracts (observations) resulted in the development of a multivariate OPLS model with one predictive component and five orthogonal ones (1 + 5 + 0). The highly active *T. scandens* extracts (i.e., 100%, 80%, 60% and 40% methanol extracts) and the less active ones (i.e., 20% methanol extracts as well as water extracts) were nicely discriminated from each other by the predictive component t [1] as shown in the scores scatter plot (Figure 4A), with the highly active extracts on the negative side and the less active extracts on the positive side. Moreover, the orthogonal component t_0_ [1] further separates the methanol extract from the other highly active extracts, which may indicate compositional differences between methanol extract and the other highly active extracts. The developed model was considered valid since the values of R^2^Y (cum) and Q^2^Y (cum) were above 0.5 (0.973 and 0.921, respectively) and the values of root mean square error of estimation (RMSEE) and root mean square error of cross validation (RMSEcv) were low (5.118 and 7.850, respectively) [4,28,29]. The Appendix A (Appendix A) displays the regression line between the observed IC_50_ values and the IC_50_ values predicted based on the developed model. A regression coefficient (R^2^) value of 0.9729 indicates high accuracy of the developed model.

### 3.3. Putative α-Glucosidase Inhibitory Metabolites

The loading column plot displayed in Figure 4B determines the metabolites responsible for the α-glucosidase inhibitory activity. The columns situated opposite to the IC_50_ (Y variable) column represent the ions (X variables) that are positively correlated with the activity [11]. The mass spectra of the ions of interest were compared to those in the NIST14 reference library, and only the metabolites with matching index more than 70% were considered [30]. In this study, 16 metabolites were identified to be responsible for the α-glucosidase inhibitory activity (Figure 5). These metabolites belong to different chemical classes, viz.: fatty acids (linoleic acid, α-linolenic acid, stearic acid, palmitic acid and its glyceryl ester; 1-monopalmitin), sterols (stigmasterol, β-sitosterol, cycloartenol and its derivative 24-methylenecycloartenol acetate), anthraquinones (emodin and its methyl ether; questin), flavanols (catechin), acyclic diterpene alcohols (phytol), fatty alcohols (1-triacontanol), in addition to α-tocopherol and 5-methoxy-8,8-dimethyl-10-(3-methyl-2-butenyl)-2*H*,8*H*-pyrano[3,2-g]chromen-2-one (Table 1). The 3D structures of the putative α-glucosidase inhibitory metabolites were further docked into the active site of the enzyme isomaltase (from *Saccharomyces cerevisiae*) crystal structure, obtained from the Protein Data Bank (PDB ID: 3A4A) using AutoDock Vina. The docking method was initially validated through control docking of the experimentally bound ligand (i.e., maltose) before proceeding to dock the putative active metabolites (Figure 6). The 2D diagrams of the best-docked conformation of each of the putative active metabolites in the active site of *Saccharomyces cerevisiae* isomaltase are provided in the Appendix A. Figure 7 shows the superimposed 3D diagram of all putative α-glucosidase inhibitory metabolites, as well as quercetin, docked into the active site of the enzyme. The predicted binding energies (indicative of the predicted affinity between the ligands and the enzyme) as well as the predicted interacting residues are shown in Table 2. The high affinity predicted between the putative α-glucosidase inhibitor metabolites and the active site of isomaltase is another supporting in silico evidence that augments the results of the GC–MS metabolomics study. 

It is noticeable that the peaks of the putative active metabolites that are common in both chromatograms are more intense in the methanol extract than in the water extract. Moreover, the peaks of the metabolites phytol (2), linoleic acid (3), α-linolenic acid (4), 1-monopalmitin (5), catechin (10), α-tocopherol (11), stigmasterol (12), β-sitosterol (13), 1-triacontanol (14), cycloartenol (15) and 24-methylenecycloartenol acetate (16) have disappeared completely from the chromatogram of the water extract. These qualitative and quantitative differences can explain the great difference observed in the α-glucosidase inhibitory activity of both extracts.

Two unsaturated fatty acids; linoleic acid (C18:2) and α-linolenic acid (C18:3) as well as two saturated ones; stearic acid (C18:0) and palmitic acid (C16:0) in addition to its glyceryl ester, 1-monopalmitin were identified by the multivariate model as α-glucosidase inhibitors. The α-glucosidase inhibitory activity of the free fatty acids has been proposed previously by similar metabolomics studies [5] and has also been verified by an in vitro enzymatic assay in other studies [31,32,33,34,35]. Investigation of the docking results showed that the carboxylic group of the fatty acids forms 1 to 3 hydrogen bonds. The number of possible hydrogen bonds increased in case of 1-monopalmitin due to the free hydroxyl groups of glycerol which represent additional sites for hydrogen bonding. Moreover, the long hydrophobic tail of the fatty acids forms numerous (11 to 14) hydrophobic interactions. The binding energies of these metabolites ranged from −6.1 to −6.5, indicating moderate affinities. 

Another class of identified α-glucosidase inhibitory metabolites was sterols, represented by stigmasterol, β-sitosterol, cycloartenol and its derivative 24-methylenecycloartenol acetate. The α-glucosidase inhibitory activity has already been reported for stigmasterol and β-sitosterol [5,36,37], whereas it is the first time to report such activity to cycloartenol and 24-methylenecycloartenol acetate as per our knowledge. The hydroxyl group at C3 of the steroid nucleus is the only polar site in the structures of the aforementioned phytosterols. This hydroxyl group formed a hydrogen bond with the residue Pro312 in the predicted ligand–enzyme complexes of all the identified sterols, except for 24-methylenecycloartenol acetate, where this hydroxyl group was involved in the ester bond (i.e., not a free hydroxyl). By contrast, the hydrophobic interactions were dominant in the ligand–enzyme complexes of these sterols due to their non-polar structures. The binding energies of these complexes ranged from −8.8 to −9.8 kcal/mol, indicating high affinities of these phytosterols towards the active site of the enzyme.

Anthraquinones (also called 9,10-anthracenediones) is a widely distributed class of phenolic plant secondary metabolites. Natural as well as synthetic anthraquinones are known for their laxative, antioxidant, antibacterial, antifungal, antiviral, anti-inflammatory and anticancer activities [38,39]. Many studies have reported the significant α-glucosidase inhibitory potential of different anthraquinones [40,41,42,43]. Different anthraquinone metabolites usually show similar biological activities, and such phenomenon has led some researchers to claim that these activities are attributed to the basic anthraquinone nucleus and the difference in strength between these compounds is due to differences in substitutions. In this study, emodin and its 8-methyl ether, questin, have been identified as α-glucosidase inhibitory metabolites. These compounds have been previously isolated from the ethyl acetate fraction of the methanol extract of *Cassia obtusifolia* seeds and assayed for their α-glucosidase inhibitory potential. Emodin showed high activity with an IC_50_ value of 1.02 µg/mL, while questin showed much less activity with an IC_50_ of 136.19 µg/mL [40]. The molecular docking results showed the importance of the hydroxyl groups of both anthraquinones in binding to the active site of α-glucosidase, where their oxygen atoms formed hydrogen bonds with Arg315, Pro312, and Tyr158. Moreover, the oxygen atom of the ketone group at C10 also formed a hydrogen bond with His280. Furthermore, questin showed more possibilities for hydrophobic interactions than emodin, with the former having 6 hydrophobic contacts (with Phe314, Leu313, Glu411, Gln279, Glu277 and Phe303 residues), while the latter having only 3 hydrophobic contacts (with Phe314, Phe303 and Gln279 residues). This higher number of possible hydrophobic interactions could be attributed to the extra methyl group of questin. This significant number of possible interactions has led to high affinity of the two compounds towards the active site of the enzyme, as reflected by the low binding energies (−8.2 and −8.3 kcal/mol for emodin and questin, respectively). Catechin, phytol, α-tocopherol, 5-methoxy-8,8-dimethyl-10-(3-methyl-2-butenyl)-2*H*,8*H*-pyrano[3,2-g]chromen-2-one as well as 1-triacontanol were also identified by the multivariate model as α-glucosidase inhibitors. The α-glucosidase inhibitory activity has already been reported to the first 3 metabolites either theoretically via similar metabolomics studies or experimentally via an in vitro enzyme assay [5,11,44]. To our knowledge, it is the first time the α-glucosidase inhibitory activity to be reported for 5-methoxy-8,8-dimethyl-10-(3-methyl-2-butenyl)-2*H*,8*H*-pyrano[3,2-g]chromen-2-one and 1-triacontanol. Molecular docking results showed that α-tocopherol, 5-methoxy-8,8-dimethyl-10-(3-methyl-2-butenyl)-2*H*,8*H*-pyrano[3,2-g]chromen-2-one and catechin exhibit high affinity towards the active pocket of the enzyme as indicated by their low binding energies (−9.3, −8.2 and −8.1 kcal/mol, respectively), whereas phytol showed moderate affinity (−6.8 kcal/mol). By investigating the interactions in the best docked conformation–enzyme complexes for these metabolites, it was noticeable that both catechin and 5-methoxy-8,8-dimethyl-10-(3-methyl-2-butenyl)-2*H*,8*H*-pyrano[3,2-g]chromen-2-one formed only a single hydrogen bond (with Arg315 and His280, respectively) compared to 2 hydrogen bonds for α-tocopherol (with Arg315 and Asp307) and phytol (with Arg446 and Asp352).

## 4. Conclusions

*Tetracera scandens* is a traditional antidiabetic herb, widely distributing in Southeast Asian countries. This study demonstrated the significant α-glucosidase inhibitory potential of the methanolic extracts of *T. scandens* leaves. Moreover, an OPLS multivariate model was developed to correlate the mass spectral data of the prepared extracts to the corresponding α-glucosidase inhibitory IC_50_ values. GC–MS based profiling of the active metabolites led to the characterization of 16 metabolites as α-glucosidase inhibitory compounds, viz.: palmitic acid, phytol, linoleic acid, α-linolenic acid, 1-monopalmitin, 5-methoxy-8,8-dimethyl-10-(3-methyl-2-butenyl)-2*H*,8*H*-pyrano[3,2-g]chromen-2-one, stearic acid, questin, emodin, catechin, α-tocopherol, stigmasterol, β-sitosterol, 1-triacontanol, cycloartenol and 24-methylenecycloartenol acetate. Furthermore, a molecular docking study was carried out to predict the binding affinities and the possible interactions of the ligand–enzyme complexes. The putative α-glucosidase inhibitory metabolites showed moderate to high affinities toward the active pocket of *Saccharomyces cerevisiae* isomaltase as indicated by their predicted binding energies (−5.9 to −9.8 kcal/mol). Conclusively, this study demonstrated the α-glucosidase inhibitory potential of the hydromethanolic extracts of *T. scandens* leaves and determined the metabolites that elicited this activity through the GC–MS-based metabolite profiling approach.

## Figures and Tables

**Figure 1 biomolecules-10-00287-f001:**
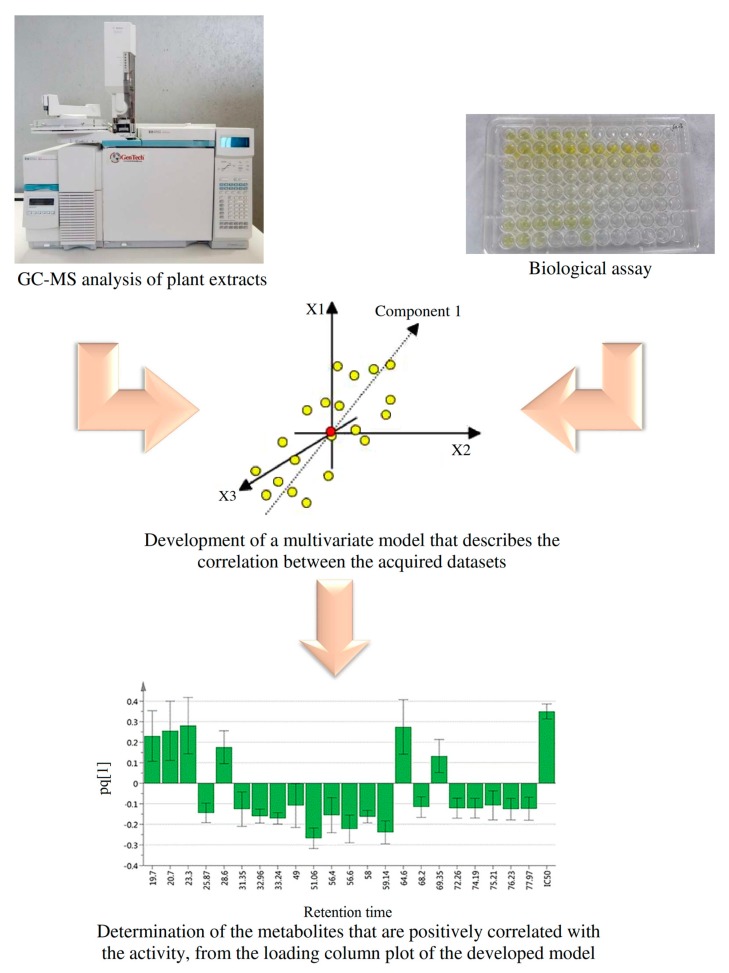
Typical procedure of the gas chromatography–mass spectrometry (GC–MS) metabolomics studies aiming to characterize the active plant metabolites.

**Figure 2 biomolecules-10-00287-f002:**
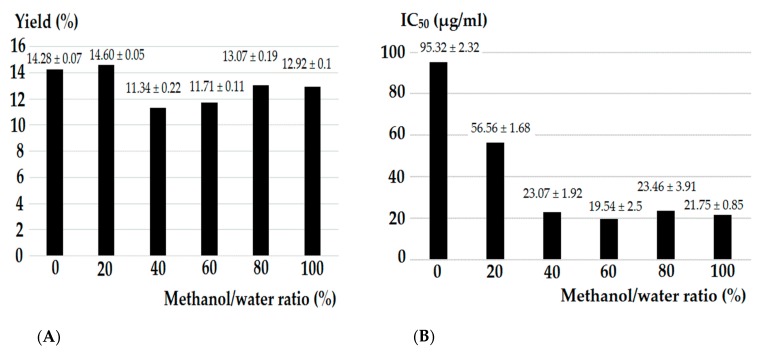
Extraction yield (%) (**A**) and α-glucosidase inhibitory activity (IC_50_; µg/mL) (**B**) of *T. scandens* leaves hydromethanolic extracts. Values expressed as the mean ± standard deviation of six replicates. IC_50_ value of the positive control (Quercetin) was 3.29 ± 0.48 µg/mL.

**Figure 3 biomolecules-10-00287-f003:**
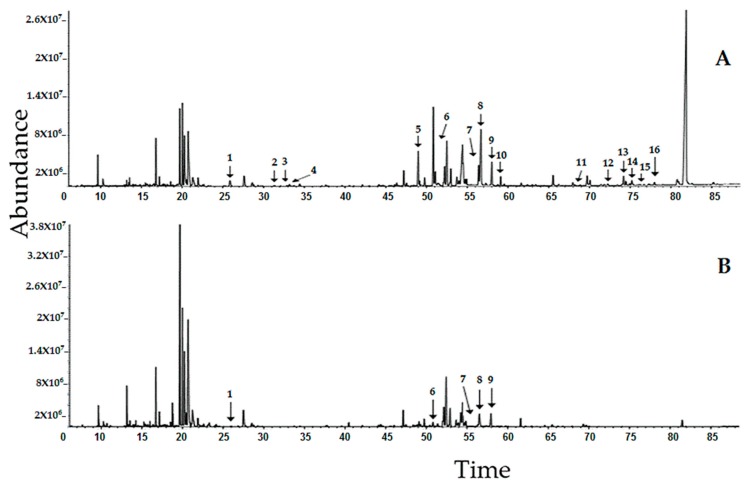
Representative chromatogram of the methanol extract (**A**) and water extract (**B**) of *T. scandens* leaves. Peak assignments: **1**—palmitic acid, **2**—phytol, **3**—linoleic acid, **4**—α-linolenic acid, **5**—1-monopalmitin, **6**—5-methoxy-8,8-dimethyl-10-(3-methyl-2-butenyl)-2*H*,8*H*-pyrano[3,2-g] chromen-2-one, **7**—stearic acid, **8**—questin, **9**—emodin, **10**—catechin, **11**—α-tocopherol, **12**—stigmasterol, **13**—β-sitosterol, **14**—1-triacontanol, **15**—cycloartenol, **16**—24-methylenecycloartenol acetate.

**Figure 4 biomolecules-10-00287-f004:**
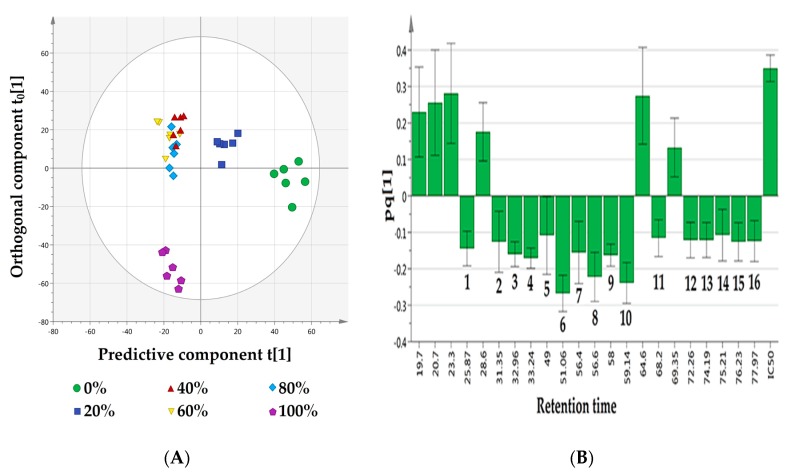
(**A**) Scores scatter plot of the established orthogonal partial least squares (OPLS) model for *T. scandens* leaves extracts, showing the highly active extracts on the negative side of the predictive component t[1], while the less active extracts on its positive side. (**B**) Loading column plot of the developed OPLS model. Assignments in **B**: **1**—palmitic acid, **2**—phytol, **3**—linoleic acid, **4**—α-linolenic acid, **5**—1-monopalmitin, **6**—5-methoxy-8,8-dimethyl-10-(3-methyl-2-butenyl)-2*H*,8*H*-pyrano[3,2-g] chromen-2-one, **7**—stearic acid, **8**—questin, **9**—emodin, **10**—catechin, **11**—α-tocopherol, **12**—stigmasterol, **13**—β-sitosterol, **14**—1-triacontanol, **15**—cycloartenol, **16**—24-methylenecycloartenol acetate.

**Figure 5 biomolecules-10-00287-f005:**
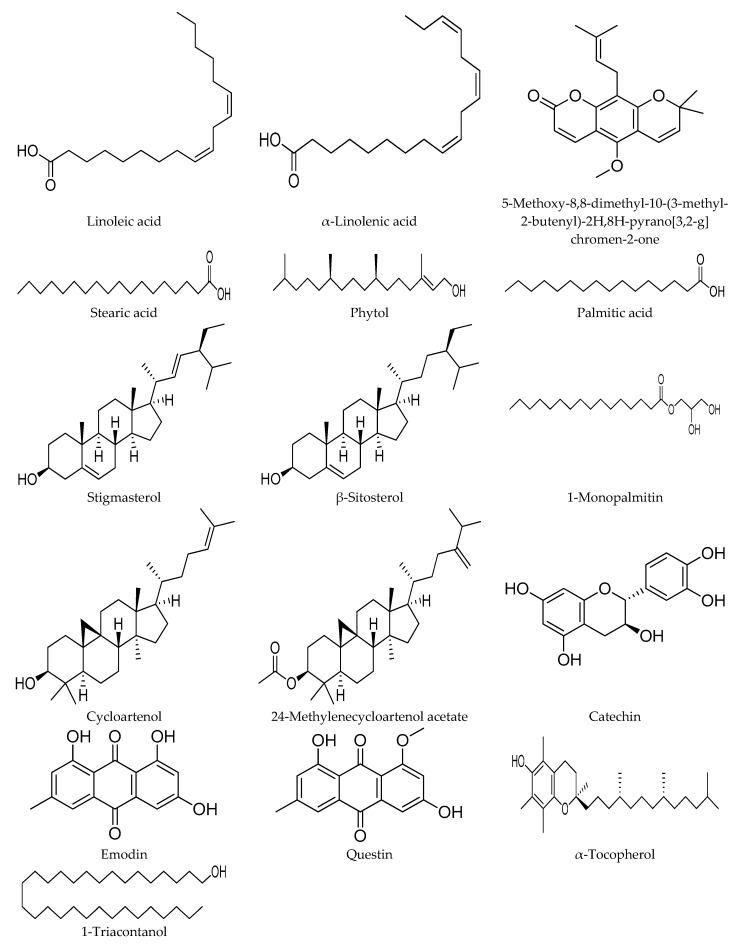
Putative α-glucosidase inhibitory metabolites.

**Figure 6 biomolecules-10-00287-f006:**
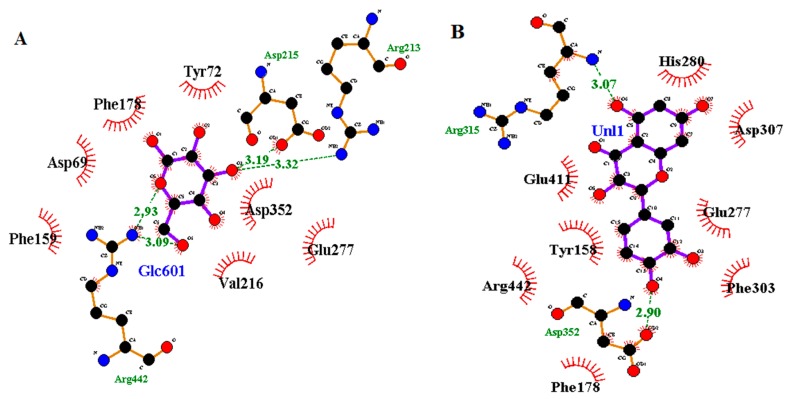
The 2D diagram of the control docking (**A**) and the positive control (quercetin) docking (**B**), with the red lashes representing the residues involved in hydrophobic interactions, while the green dotted lines represent the hydrogen bonds.

**Figure 7 biomolecules-10-00287-f007:**
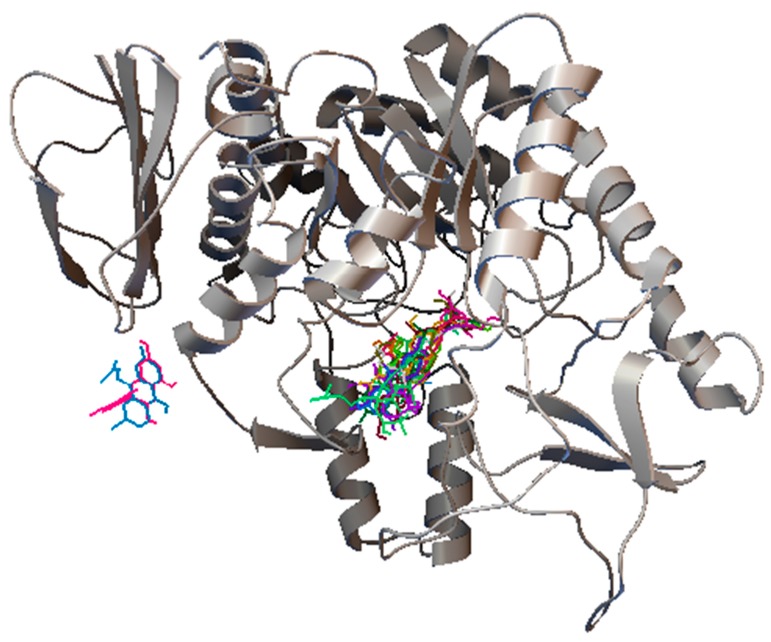
The superimposed 3D diagram showing all the putative active metabolites as well as quercetin, docked into the active site of *Saccharomyces cerevisiae* isomaltase. The 2 metabolites away from the remaining overlapped metabolites are 5-methoxy-8,8-dimethyl-10-(3-methyl-2-butenyl)-2*H*,8*H*-pyrano[3,2-g] chromen-2-one and catechin.

**Table 1 biomolecules-10-00287-t001:** α-Glucosidase inhibitory metabolites identified from *T. scandens* leaves extracts through GC–MS metabolomics.

No	RT (min)	Matching Index	Probability%	Molecular Weight	Molecular Formula	Putative Metabolite
1	25.87	94.3	98.4	256.43	C_16_H_32_O_2_	Palmitic acid
2	31.35	87.5	92.9	296.53	C_20_H_40_O	Phytol
3	32.96	76.6	92.7	280.45	C_18_H_32_O_2_	Linoleic acid
4	33.24	73.9	59.6	278.43	C_18_H_30_O_2_	α-Linolenic acid
5	49	90.5	94.6	330.5	C_19_H_38_O_4_	1-Monopalmitin
6	51.06	79	77.8	326.39	C_20_H_22_O_4_	5-Methoxy-8,8-dimethyl-10-(3-methyl-2-butenyl)-2*H*,8*H*-pyrano[3,2-g] chromen-2-one
7	56.4	88.1	86.8	284.48	C_18_H_36_O_2_	Stearic acid
8	56.6	75.7	45.8	284.26	C_16_H_12_O_5_	Questin
9	58	83.3	69.4	270.24	C_15_H_10_O_5_	Emodin
10	59.14	81.4	76.4	290.27	C_15_H_14_O_6_	Catechin
11	68.2	88.3	59.1	430.71	C_29_H_50_O_2_	α-Tocopherol
12	72.26	76.1	69.8	412.7	C_29_H_48_O	Stigmasterol
13	74.19	93.7	97.7	414.71	C_29_H_50_O	β-Sitosterol
14	75.21	92.8	85.6	438.82	C_30_H_62_O	1-Triacontanol
15	76.23	80.7	89.1	426.72	C_30_H_50_O	Cycloartenol
16	77.97	83.8	77.8	482.79	C_33_H_54_O_2_	24-Methylenecycloartenol acetate

RT = retention time, C = carbon, H = hydrogen, O = oxygen.

**Table 2 biomolecules-10-00287-t002:** Molecular docking results of the identified active metabolites as well as the positive control (quercetin) onto the active site of *Saccharomyces cerevisiae* isomaltase. Common interacting residues within the same metabolite class (i.e., fatty acids, anthraquinones and phytosterols) are highlighted by bold and/or underline.

Docked Metabolite	Predicted Binding Energy (Kcal/mol)	Residues Involved in Hydrogen Bonding	Residues Involved in Hydrophobic Interactions
Maltose (control docking)	−6	Arg213, Arg442, Asp215	Tyr72, Phe178, Asp69, Phe159, Asp352, Glu277, Val216
Quercetin (positive control)	−8.8	Arg315, Asp352	His280, Asp307, Glu277, Phe303, Phe178, Arg442, Tyr158, Glu411
Palmitic acid	−6.1	Gln353	**Arg315**, Arg442, **Phe303**, **Asp352**, **Glu411**, **Tyr158**, Glu277, **Tyr72**, **Phe178**, Phe159, Val216, Asp215, Asp69, His351
Phytol	−6.8	Arg446, Asp352	Arg315, Phe303, Gln353, Arg442, His351, Asp69, Asp215, Tyr72, Glu277, Val216, Phe178, Glu411, Gln279, Tyr158
Linoleic acid	−6.4	Asp215, His351	**Glu411**, Val216, **Phe178**, Gln279, **Phe303**, **Arg315**, His280, Asp307, Gln353, **Tyr158**, Arg442, **Asp352**, **Tyr72**
α-Linolenic acid	−6.5	Arg213, Glu277	Gln279, Phe59, Arg442, **Glu411**, **Phe178**, **Tyr158**, Val216, Asp215, **Tyr72**, Asp307, **Asp352**, Gln353, **Phe303**, **Arg315**
1-Monopalmitin	−5.9	Gln182, His112, Asp69, Arg442,	Asp215, **Tyr72**, **Phe178**, **Tyr158**, Phe159, **Glu411**, Asp307, **Arg315**, **Asp352**, **Phe303**, Gln279
5-Methoxy-8,8-dimethyl-10-(3-methyl-2-butenyl)-2*H*,8*H*-pyrano[3,2-g] chromen-2-one	−8.2	His 280	Phe159, Glu411, Tyr158, Phe178, Arg442, Arg315, Phe303, Asp307
Questin	−8.3	**Pro312**, **His280**, **Tyr158**, **Arg315**	**Gln279**, Glu277, Glu411, Leu313, **Phe314**, **Phe303**
Emodin	−8.2	**Arg315**, **Pro312**, **His280**, Glu411, **Tyr158**	**Gln279**, **Phe303**, **Phe314**
Catechin	−8.4	Arg315	His280, Asp307, Glu277, Phe303, Asp352, Arg442, Phe178, Tyr158, Glu411
α-Tocopherol	−8.3	Arg315, Asp307	Ser311, Pro312, Asp242, Tyr158, Glu411, Arg442, Phe159, Phe178, Asp215, His112, Val216, Tyr72, Asp69, Gln279, Phe303
Stigmasterol	−9.1	Ser311, Pro312	Asp215, Arg442, Asp69, Tyr72, His351, Val216, Gln182, Glu411, Phe178, Asp352, Glu277, His112, Tyr158, Gln279, Phe159, Arg315, Phe303, Leu313, Phe314, Asp307
β-Sitosterol	−8.8	Pro312	Leu313, Phe314, Tyr158, Asp69, Asp352, Glu411, Arg442, Tyr72, His351, His112, Arg213, Asp215, Phe303, Phe178, Glu277, Gln279, Arg315
Cycloartenol	−9.8	Pro312	Asp242, Ser240, Tyr158, Glu411, Glu277, Arg442, Asp215, Phe178, Asp352, Asp69, Arg315, Gln279,
24-Methylenecycloartenol acetate	−8.8	Arg442	Asp352, Phe178, Glu411, Phe159, Gln279, His280, Thr310, Arg315, Asp307, Pro312, Phe303, Tyr158, Glu277

Docking of stearic acid and 1-triacontanol was not possible due to unavailability of the 3D structures of the metabolites.

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
