# Peer review of "Investigation of α-Glucosidase Inhibitory Metabolites from Tetracera scandens Leaves by GC–MS Metabolite Profiling and Docking Studies"

_biomolecules, 2020, doi:10.3390/biom10020287_

Round 1
Reviewer 1 Report
In the Manuscript titled “Investigation of α-glucosidase inhibitory metabolites from Tetracera scandens leaves by GC-MS metabolite profiling and docking studies”, authors have brought out some useful information on their studies. But still the rationale behind the study needs to be simplified and explained in understandable way (as a figure or flow chart will enhance reader understanding). I recommend this manuscript for major revision before consideration for publication.
Points to consider:
Why the overall yield of the extract is low? The introduction needs to highlight various other similar studies carried out and their observations. Explain in detail about the OPLS model What is the significance of the error in figure 6? Add details in the figure caption. What is the pH used to generate the PDB2PQR? Explain in detail about the parameters used for Molecular docking experiments. How many times were the AutoDock experiments repeated? Add a figure of the docking profile with protein and the overlap image of all possible docked metabolites Give the significance of similar and dissimilar residues from the binding studies in table 2 by highlighting … like text in bold Generate a figure showing the best-docked pose of each molecule with the protein in a bound formAuthor Response
Thank you for the valuable comments. Below are our responses as per the constructive comments made by the reviewer 1:
|
Authors have brought out some useful information on their studies. But still the rationale behind the study needs to be simplified and explained in understandable way (as a figure or flow chart will enhance reader understanding). |
We added a flow chart in the introduction to simplify the concept |
|
Why the overall yield of the extract is low? |
The extraction yield in our study (11.34% to 14.60%) is comparable to the yield reported from the leaves of the same plant in previous studies: 10% to 13.69% (Umar et al 2010, doi:10.1016/j.jep.2010.06.016), 10% (Thanh et al, doi:10.1016/S2221-1691(15)30009-5).
|
|
The introduction needs to highlight various other similar studies carried out and their observations. |
Some similar previous studies were already highlighted in brief in the introduction (2nd paragraph). But we added more about their findings as per your recommendation |
|
Explain in detail about the OPLS model |
We explained more about the concept behind the PLS model and its recent extension; the OPLS model, in the introduction (2nd paragraph). We tried to simplify this concept to make it easier to understand for the readers who are not familiar with multivariate statistics |
|
What is the significance of the error in figure 6? |
The OPLS model was developed based on 2689 X-variables. Before the determination of the ions (i.e. X variables) contributing to the activity (i.e. Y variable), this huge number needed to be shortlisted. Initially, the variable importance for the projection (VIP) plot was developed. This plot summarizes the importance of the variables both to explain X and to correlate to Y. the VIP plot is sorted from high to low, and shows confidence intervals for the VIP values at the 95% level. VIP values larger than 1 indicate important X-variables. So, we excluded all X variables with VIP values less than 1. Then the loadings column plot was developed. Thereafter, the columns (representing the X variables with VIP values higher than 1) were observed one by one to investigate the error bars. All columns with too large error bars (i.e. the error bar is extending out of the column in both up and down directions) were further excluded (indicate non-significant difference). Now we ensured that the remaining columns (i.e. X variables) were considered statistically significant. |
|
Add details in the figure caption. |
Done |
|
What is the pH used to generate the PDB2PQR? |
We added the pH value (pH = 6.5) |
|
Explain in detail about the parameters used for Molecular docking experiments. |
We explained in more detail the procedure followed for molecular docking. |
|
How many times were the AutoDock experiments repeated? |
The docking job was done once, but after your comment we repeated it more 2 times on different occasions and the results of the 3 times are summarized in the table below |
|
Add a figure of the docking profile with protein and the overlap image of all possible docked metabolites |
Done (figure 7) |
|
Give the significance of similar and dissimilar residues from the binding studies in table 2 by highlighting … like text in bold |
Common interacting residues within the same metabolite class (i.e. fatty acids, anthraquinones and phytosterols) are highlighted by bold and/or underline |
|
Generate a figure showing the best-docked pose of each molecule with the protein in a bound form |
It would be too large to be published in the article, so we added it as supplementary data (File S2) |
|
Docked metabolite |
Predicted binding energy (Kcal/mol) |
||
|
Trial 1 |
Trial 2 |
Trial 3 |
|
|
Maltose (control docking) |
-6 |
-6 |
-6 |
|
Quercetin (positive control) |
-8.8 |
-8.8 |
-8.8 |
|
Palmitic acid |
-6.1 |
-5.9 |
-5.9 |
|
Phytol |
-6.8 |
-6.6 |
-6.6 |
|
Linoleic acid |
-6.4 |
-6.3 |
-6.6 |
|
α-Linolenic acid |
-6.5 |
-6.7 |
-6.7 |
|
1-Monopalmitin |
-5.9 |
-6.1 |
-5.9 |
|
5-Methoxy-8,8-dimethyl-10- (3-methyl-2-butenyl) -2H,8H-pyrano[3,2-g] chromen-2-one |
-8.2 |
-8.2 |
-8.2 |
|
Questin |
-8.3 |
-8.3 |
-8.3 |
|
Emodin |
-8.2 |
-8.2 |
-8.2 |
|
Catechin |
-8.4 |
-8.4 |
-8.4 |
|
α-Tocopherol |
-8.3 |
-8.4 |
-8.3 |
|
Stigmasterol |
-9.1 |
-9.1 |
-9.1 |
|
β-Sitosterol |
-8.8 |
-8.4 |
-8.8 |
|
Cycloartenol |
-9.8 |
-9.8 |
-9.8 |
|
24-Methylenecycloartenol acetate |
-8.8 |
-8.8 |
-8.9 |

Reviewer 2 Report
The authors have done an interesting job for anti-α-glucosidase naturally-occurring bioactive ingredients from Tetracera scandens by integrating GC-MS-based metabolomics approach with chemometrics analysis. Totally, 16 metabolites were putatively identified by matching with NIST library and the result were partially validated by computational docking model. They have a rigorous experimental design for sample collection, data acquisition followed by data preprocessing and analysis, however, I have some suggestions for the authors to consider.
Page 5: There is no title for neither X-axis nor Y-axis in Figure 1A and B. Page 6: I suggest Figure 2 and Figure 3 can be merged as one for better visualization and comparison as they are acquired by the same platform. And it would be better if you can include the special compound name for each assigned peak in the legend but not just ask readers to find from Table 1. Page 7: Figure 4 and Figure 6 can be merged since both of them were generated by OPLS analysis. Score plot is always paired with loading plot or S-plot. Page 7: Figure 5 is mainly a supplementary validation for the prediction function of OPLS model. But the purpose of the model in your study is to identify the target peaks who contribute to the bioactivity or clustering happened in score plot but not the prediction of IC50. So I suggest to move it to Supplementary Information. Page 8, Figure 6: The label of x-axis for loading plot should be pair-identifier for the peak (X-variable), it can be represented by mass to charge ratio and Rt(m/z/Rt). However, I would recommend you including S-plot instead of loading plot. Because S-plot is specific to OPLS model and it can provide a more reliable rankling list by removing unrelated variation in the orthogonal direction. Page 4: In the data preprocessing and statistical section, please indicate which kind of scaling and normalization method you use if have applied. Different scaling and normalization methods may lead totally different results. You may refer to Proteome Res.2016, 15(8): 2595-2606 for more details. Most importantly, validation part is missing in this manuscript. No matter multivariate data analysis or docking strategy is just for narrowing down the putative candidates, biological validation is necessary for the robustness of your result if it is going to be published. What’s more, most of the metabolites identified by your study are common and commercially available, at least the top 3 of them with highest scores need to be validated by the α-glucosidase inhibitory assay.Author Response
Dear Editor-in-Chief,
Thank you for your valuable comments. Here are my responses to the reviewer 2:
|
Page 5: There is no title for neither X-axis nor Y-axis in Figure 1A and B. |
Performed as per request |
|
Page 6: I suggest Figure 2 and Figure 3 can be merged as one for better visualization and comparison as they are acquired by the same platform. |
Performed as per request |
|
And it would be better if you can include the special compound name for each assigned peak in the legend but not just ask readers to find from Table 1. |
Performed as per request |
|
Page 7: Figure 4 and Figure 6 can be merged since both of them were generated by OPLS analysis. |
Performed as per request |
|
Score plot is always paired with loading plot or S-plot. Page 7: Figure 5 is mainly a supplementary validation for the prediction function of OPLS model. But the purpose of the model in your study is to identify the target peaks who contribute to the bioactivity or clustering happened in score plot but not the prediction of IC50. So I suggest to move it to Supplementary Information. |
Performed as per request. The observed vs predicted plot is attached as a supplementary figure (Figure S1) |
|
Page 8, Figure 6: The label of x-axis for loading plot should be pair-identifier for the peak (X-variable), it can be represented by mass to charge ratio and Rt(m/z/Rt).
However, I would recommend you including S-plot instead of loading plot. Because S-plot is specific to OPLS model and it can provide a more reliable rankling list by removing unrelated variation in the orthogonal direction. |
We used XCMS (R version) for preprocessing of the mass spectral data. This software gives a specific name for each feature, based on the m/z and RT. Unlike XCMS online which mentions the RT in minutes, XCMS R package mentions the RT in seconds (e.g. M146T4450). XCMS gives 2 more columns for the m/z and RT (seconds) data individually. So, it would be confusing to the reader if we used such pair-identifier names to represent the features. Instead, we developed a new column for RT in minutes to represent the features, as we think it is more important to the reader than the m/z data.
We used the UV scaling during the multivariate analysis of this study, so it is not possible to develop the S plot for the established OPLS model (the S plot only developed with Pareto or Ctr scaling; source: SIMCA help). However, we tried to use pareto scaling of the data, just to compare the results of the S plot with the results we got from the loading column plot. We got a new OPLS model (1+2+0). The features representing the metabolites questin, emodin, 1-monopalmitin, catechin and 5-Methoxy-8,8-dimethyl-10-(3-methyl-2-butenyl)-2H,8H-pyrano[3,2-g] chromen-2-one were situated on the wing of the S plot, indicating high model influence and high reliability. Furthermore, as we go toward the vertical diagonal, most of the remaining putative metabolites were found. Unfortunately, the different extracts were not discriminated in a good way in the scores scatter plot of this new OPLS model and the R2Y (cum) and Q2Y (cum) for this model were 0.88 and 0.825, respectively. So, we think that the initial OPLS model was better (best describes the variation in X and Y as indicated by the higher R2Y and Q2Y, nice discrimination of the different extracts in the scores scatter plot, identified the same putative metabolites as the S plot of the new model did). Moreover, as per our knowledge, UV scaling is normally used for FT-IR and mass spectra, while pareto scaling is better for NMR spectra. |
|
Page 4: In the data preprocessing and statistical section, please indicate which kind of scaling and normalization method you use if have applied. Different scaling and normalization methods may lead totally different results. You may refer to Proteome Res.2016, 15(8): 2595-2606 for more details. |
We used only the UV scaling. We mentioned it as per your recommendation. |
|
Most importantly, validation part is missing in this manuscript. No matter multivariate data analysis or docking strategy is just for narrowing down the putative candidates, biological validation is necessary for the robustness of your result if it is going to be published. What’s more, most of the metabolites identified by your study are common and commercially available, at least the top 3 of them with highest scores need to be validated by the α-glucosidase inhibitory assay. |
The in-vitro α-glucosidase inhibitory potential has already been investigated previously for almost all the commercially-available standards of the putative active metabolites identified in this study, that is why we skipped this step. Palmitic acid, stearic acid, linoleic acid and linolenic acid standards showed IC50 values of 95.27, 88.22, 2.79 and 4.31 µg/mL, respectively (Indrianingsih & Tachibana, 2017). Moreover, β-sitosterol and stigmasterol were isolated from the leaves of Dillenia indica and showed 52.5 and 34.2% inhibition in α-glucosidase activity (kumar et al., 2013). Furthermore, emodin and questin were isolated from Cassia obtusifolia seeds and showed α-glucosidase inhibitory IC50 values of 1.02 and 136.19 µg/mL, respectively (Jung et al., 2017). Similarly, catechin was isolated from the leaves of Muehlenbeckia tamnifoliaa and showed α-glucosidase inhibitory IC50 of 5.5µM). |

Round 2
Reviewer 1 Report
I do recommend for publication.
Reviewer 2 Report
Thanks for the authors' effort spent on the revision of the manuscript. It has improved a lot but the most important part, the missing experiment for biological validation of the target compounds identified by chemometrics model still has not been figured out. Although the alpha-glucosidase inhibitory activity of such known compounds have been published, but it does not mean you can skip it in your own model. Generally, in this manuscript by applying the chemometrics approach, several known compounds have been identified with potential activity with no validation. A paper with very similar methodology and scenario was published 5 years ago. (J Food Sci. 2014 Jun;79(6):C1130-6. doi: 10.1111/1750-3841.12491. Epub 2014 Jun 2.) The only difference between these two manuscripts is working on different herbs but with no improvement on methodology, experimental design or significance of research (no new finding). So unfortunately, I can not accept it by comparing previously published peer-reviewed paper.
